# Monascin and Ankaflavin of *Monascus purpureus* Prevent Alcoholic Liver Disease through Regulating AMPK-Mediated Lipid Metabolism and Enhancing Both Anti-Inflammatory and Anti-Oxidative Systems

**DOI:** 10.3390/molecules26206301

**Published:** 2021-10-18

**Authors:** Jhao-Ru Lai, Ya-Wen Hsu, Tzu-Ming Pan, Chun-Lin Lee

**Affiliations:** 1Department of Life Science, National Taitung University, 369, Section 2, University Rd., Taitung 95092, Taiwan; ruth790411@hotmail.com; 2R&D Division, SunWay Biotechnology Company Limited, Taipei 10617, Taiwan; hyw@sunway.cc; 3Department of Biochemical Science and Technology, National Taiwan University, Taipei 10617, Taiwan

**Keywords:** *Monascus purpureus*, monascin, ankaflavin, alcoholic liver diseases, alcoholic fatty liver

## Abstract

Alcohol metabolism causes an excessive accumulation of liver lipids and inflammation, resulting in liver damage. The yellow pigments monascin (MS) and ankaflavin (AK) of *Monascus purpureus*-fermented rice were proven to regulate ethanol-induced damage in HepG2 cells, but the complete anti-inflammatory and anti-fatty liver mechanisms in the animal model are still unclear. This study explored the roles of MS and AK in improving alcoholic liver injury. MS and AK were simultaneously fed to evaluate their effects and mechanisms in C57BL/6J mice fed the Lieber–DeCarli liquid alcohol diet for 6 weeks. The results indicated that MS and AK significantly reduced the serum aspartate aminotransferase and alanine aminotransferase activity, as well as the total liver cholesterol and triglyceride levels. The histopathological results indicated that MS and AK prevented lipid accumulation in the liver. MS and AK effectively enhanced the activity of antioxidant enzymes and reduced the degree of lipid peroxidation; AK was particularly effective and exhibited a superior preventive effect against alcoholic liver injury and fatty liver. In addition to inhibiting the phosphorylation of the MAPK family, MS and AK directly reduced TNF-α, IL-6, and IL-1β levels, thereby reducing NF-κB and its downstream iNOS and COX-2 expressions, as well as increasing PPAR-γ, Nrf-2, and HO-1 expressions to prevent liver damage. MS and AK also directly reduced TNF-α, IL-6, and IL-1β expression, thereby reducing the production of NF-κB and its downstream iNOS and COX-2, and increasing PPAR-γ, Nrf-2, and HO-1 expressions, preventing alcohol damage to the liver.

## 1. Introduction

The initial stage of liver disease is liver damage. Chronic liver damage can lead to fibrosis, liver failure, and liver cancer [1]. Causes of liver damage include fatty liver, viral infection, and alcohol use. These causal factors can interact and accelerate liver damage. The metabolism of alcohol in the liver has three paths, namely alcohol dehydrogenase (ADH) activity, the microsomal ethanol-oxidizing system of cytochrome P450 2E1 (CYP2E1), and catalase (CAT) activity [2,3]. However, an imbalance in the redox state of liver cells and the generation of free radicals during the metabolic process have been the main research directions for alcoholic liver injury [4,5]. In addition, alcohol affects the key products of fat synthesis and degradation balance [6], and induces lipid synthase [7], resulting in alcoholic fatty liver.

*Monascus* spp. has been used in food and medicine for centuries and in recent years it has gradually developed a reputation as a health food. The reported effects of *Monascus purpureus*-fermented rice (red mold rice) include enhancing the activity of antioxidant enzymes in the liver of mice, inhibiting the expression of inflammation-related factors, and reducing the damage caused by alcohol to liver cells [8]. In recent years, the secondary metabolites of red mold rice have been studied to further explore their functional compounds. Among them, the yellow pigments monascin (MS) and ankaflavin (AK) are proven as the functional compounds for the prevention of cardiovascular disease, fatty liver, and lipogenesis (Figure 1) [9,10,11]. Furthermore, both MS and AK can achieve anti-inflammatory effects by reducing inflammatory factor levels [12,13]. Regarding alcoholic liver disease, MS and AK were proven to regulate ethanol-induced peroxisome proliferator-activated receptor-γ and sterol regulatory element-binding transcription factor-1 expression in HepG2 cells [14]. However, the regulation mechanisms mediated by MS and AK in the ethanol diet-induced livery injury and fatty liver animal model are still unclear. In the current study, the abilities of the anti-inflammatory substances MS and AK to improve alcoholic liver injury were explored. C57BL/6J male mice were fed daily the Lieber–DeCarli diet to induce alcoholic liver injury while they were also fed various concentrations of the test substances. After 6 weeks, the effects of the substances on serum liver function indicators as well as on triglyceride (TG) and total cholesterol (TC) levels were observed, and histopathological section-staining was performed to assess the degree of lipid accumulation in the liver. The degree of liver tissue inflammation (measured through the performance of cytokines, inflammatory factor regulation, and anti-inflammatory factors) was also explored using western blotting. In addition, the protein expression levels of 5’ AMP-activated protein kinase (AMPK), sterol regulatory element-binding protein 1 (SREBP-1), acetyl-CoA carboxylase (ACC), and peroxisome proliferator-activated receptor-α (PPAR-α), which are related to lipid metabolism and biosynthesis, were analyzed to explore possible mechanisms of MS and AK in preventing alcoholic liver injury and fatty liver.

## 2. Results

### 2.1. Effects of Monascin and Ankaflavin on Body Weight and Relative Liver to Body Weight Ratio in Ethanol-induced Liver Injury Mice

In this trial, C57BL/6J male mice were fed with a normal (NOR) or a Lieber–DeCarli ethanol (EtOH) diet to induce alcoholic liver injury. Silymarin (SL, 200 mg/kg) was fed by tube to the positive control group and MS-L or -H (0.615 mg/kg or 3.075 mg/kg), or AK-L or -H (0.3075 mg/kg or 1.5375 mg/kg) was fed to the experimental groups. The experimental period was 6 weeks. The ratios of liver weight to body weight are presented in Figure 2. The liver weight–body ratio of the NOR (3.33%), MS (3.71% and 3.75%, low vs. high dose), and AK (3.68% and 3.57%, low vs. high dose) test groups were significantly lower than that of the EtOH group (4.14%, *p* <0.001). This result indicated that *Monascus purpureus*–produced MS and AK were able to repress the increase of the liver weight to body weight ratio caused by alcohol.

### 2.2. Effects of Monascin and Ankaflavin on Serum AST, ALT, and ALP Activities

As Table 1 indicates, alcohol caused significant increases in serum liver function indicator levels (*p  <* 0.05), representative of liver damage. Although SL in the positive control group effectively reduced the levels of aspartate aminotransferase (AST) and alanine aminotransferase (ALT, *p  <* 0.05), no significant difference was observed in the hepatobiliary function index level of alkaline phosphatase (ALP; *p* > 0.05). The MS-L group exhibited the same effect trend as the SL group but MS-H effectively reduced the values for all three indicators (*p <* 0.05) such that they were close to those of the NOR group. AK at low doses reduced AST, ALT, and ALP activity by 17.4%, 23%, and 10.4%, respectively (*p <* 0.01), whereas at high doses, it significantly reduced AST activity (*p <* 0.001). According to these results, MS and AK had the ability to reduce liver function indicator levels at low doses and thus reduced alcohol-induced liver damage.

### 2.3. Effects of Monascin and Ankaflavin on Serum and Hepatic TG and TC Contents

In this study, the content of TC and TG in serum and the liver were measured to investigate whether an alcohol diet caused lipid accumulation in the liver and formed fatty liver. The results are presented in Table 2. Although the serum TC of the EtOH group exhibited a tendency to increase, it was not significantly different from that of the NOR group (*p*  >  0.05). However, the TG content of the EtOH group was significantly higher by 16.72% (*p* <  0.01). In the positive control group given SL, serum TG was significantly lower (*p* <  0.05) but the decrease in TC concentration was not significant. In the MS and AK groups, both high and low doses reduced serum TC and TG concentrations (*p* <  0.05).

The thanol diet significantly increased TC and TG content in the liver by 9.55% (*p* <  0.001) and 36.21% (*p* <  0.001), respectively. In the SL, MS, and AK groups, TC and TG were prevented from accumulating in the liver (*p* <  0.001). AK had the largest effect. These results indicate that the test substances reduced the content of TC and TG, inhibited the lipid synthesis induced by alcohol, and reduced the accumulation of lipids in the liver.

### 2.4. Effects of Monascin and Ankaflavin on Hepatic Pathological Changes

After sacrifice, liver tissue sections were collected for hematoxylin and eosin-staining to observe pathological changes. As evident in the slice images in Figure 3, the internal structure of the NOR tissue liver cells were clear, evenly distributed, and arranged completely, without vacuoles or cell infiltration. In the slices from the EtOH group, the liver cells were filled with ballooning degeneration and arranged irregularly; moreover, cell infiltration around the blood vessels due to inflammation was observed. The tissues from the SL, MS-L, MS-H, AK-L, and AK-H groups all approached the NOR group pattern, and no serious accumulation of fat was observed, indicating that MS and AK can prevent both alcoholic fatty liver and the occurrence of liver damage.

### 2.5. Effects of Monascin and Ankaflavin on the Lipid Peroxidation and Activities of Antioxidant Enzymes in the Liver 

As shown in Figure 4, the malondialdehyde (MDA) concentration of the EtOH group was 51% higher than that of the NOR group (*p* <  0.001), indicating that alcohol caused severe lipid peroxidation in the liver. Compared with the EtOH group, MDA levels in the SL, MS-L, MS-H, AK-L, and AK-H groups were significantly reduced by 58.32% (*p* <  0.001), 61.43% (*p* <  0.001), 61.49% (*p* <  0.001), 74.26% (*p* <  0.001), and 78.74% (*p* <  0.001), respectively, with levels close to those in the NOR group. MS and AK exhibited significant effects in improving the liver lipid peroxidation caused by alcohol at low doses. 

Studies have concluded that alcohol can affect the activity of antioxidant enzymes in the body [15], cause metabolic-related diseases, and induce alcoholic liver disease [16,17]. This study analyzed the activity of glutathione peroxidase (GPx), glutathione reductase (GRd), catalase (CAT), and superoxide dismutase (SOD) in the liver, and explored the effects of MS and AK on the regulation of antioxidant enzyme activity. As Table 3 suggests, the data for the EtOH and NOR group were significantly different (*p*  <  0.001). Enzyme activity was significantly inhibited due to the influence of alcohol. Among enzymes, CAT activity was decreased by 22.37%, SOD activity was decreased by 41.45%, GPx activity was decreased by 24.18%, and GRd activity was decreased by 15.89 %. For the MS groups, the MS-L group exhibited significantly higher CAT (*p*  <  0.001) and GRd (*p*  <  0.05) activity, but high doses of MS were necessary for the significantly increased SOD (*p*  <  0.01) and GRd (*p*  <  0.05) activity. Increased CAT (*p* <  0.001), SOD (*p*  <  0.01), and GRd (*p*  <  0.01) activity was observed with low doses of AK, and the difference was more significant for GRd (*p*  <  0.001) at high doses. These results confirmed that MS and AK effectively enhanced the activity of antioxidative enzymes decreased by alcohol. 

### 2.6. Effects of Monascin and Ankaflavin on the Regulation of Hepatic Inflammatory Factor Expression

The effects of alcohol on the protein expression of inflammatory factors in liver tissue are depicted in Figure 5A. In Figure 5B–E, compared with the NOR group, TNF-α, IL-1β, and IL-6, NF-κB protein expression levels were increased by 5.79 times (*p* < 0.001), 4.08 times (*p* < 0.001), 2.91 times (*p* < 0.001), and 2.06 times (*p* < 0.001), respectively. Low doses of MS or AK reduced TNF-α expression levels by 45.32% (*p*  <  0.01) or 54.02% (*p*  <  0.001); those of IL-1β by 31.04% (*p*  <  0.001) or 36.38% (*p*  <  0.001); IL-6 by 32.83% (*p*  <  0.001) or 35.57% (*p*  <  0.001); and NF-κB by 15.99% (*p*  <  0.01) or 22.6% (*p*  <  0.001), respectively. These results indicate that feeding mice MS or AK has a significant effect on the inhibition of the expression of inflammatory factors, with a dose-dependent response.

Figure 5F,G presents the results for iNOS and COX-2 protein expression. The liver iNOS content in the EtOH group was significantly higher, at 235.19%, than that of the NOR group (*p*  <  0.001). However, both the MS-L and AK-L groups had significantly lower iNOS expression levels, specifically 23.74% (*p*  <  0.05) and 28.04% (*p*  <  0.01) lower, respectively, with the most significant effect at high doses (*p*  <  0.001). In addition to increasing iNOS expression, alcohol increased COX-2 (*p*  <  0.001) levels. Both the MS-L and AK-L groups had significantly reduced iNOS and COX-2 expression levels (*p*  <  0.001). Although alcohol greatly increased inflammation-related protein content in the liver, MS and AK reduced inflammation factors at low doses and the effect at high doses was superior to that of SL.

Alcohol increases intestinal permeability and initiates the phosphorylation of ERK1/2 and p38, which in turn causes inflammation. Therefore, we explored whether MS and AK achieve anti-inflammatory effects through regulation of ERK1/2 and p38. Figure 5H–K presents the results for the measurement of ERK1/2 protein expression. The test substances did not directly regulate the total amount of ERK1/2 but did prevent the production of downstream inflammatory factors by inhibiting its phosphorylation. pERK1/2 content in the EtOH group was 4.18 times that of the NOR group (*p*  <  0.001). MS-L and AK-L decreased pERK1/2 content by 17.28% (*p*  <  0.05) and 37.57% (*p*  <  0.001), respectively. Low-dose AK demonstrated a reduction effect similar to that of high-dose MS. Long-term alcohol intake also significantly increased (an increase of 5.99 times) p-p38 expression levels; however, feeding low doses of MS or AK significantly reduced p-p38 expression by 21.66% (*p*  <  0.01) and 48.13% (*p*  <  0.001), respectively. These results signified that MS and AK reduced the phosphorylation of ERK1/2 and p38 to inhibit alcohol-induced inflammation. 

### 2.7. Effects of Monascin and Ankaflavin on The Regulation of Hepatic Anti-Inflammatory Factor Expression 

When inflammation occurs in the liver, anti-inflammatory factors such as PPAR-γ, Nrf-2, and HO-1 can reduce the inflammatory response [18]. As the results shown in Figure 6A,B indicate, alcohol reduced the levels of anti-inflammatory protein PPAR-γ in the livers of the mice by 54.95% (*p* < 0.001), whereas the values in the SL group, MS-L group, and AK-L group were similar to that in the EtOH group. PPAR-γ expression levels were increased by 1.56 times (*p* < 0.01), 1.39 times (*p* < 0.05), and 1.44 times (*p* < 0.05). High doses of MS and AK further elevated the expression of PPAR-γ (*p* < 0.001). These results prove that MS and AK prevented the production of inflammatory factors by increasing PPAR-γ, thus reducing the inflammation caused by alcohol.

When oxidative stress occurs in the body, Nrf-2 is dissociated into the nucleus to produce numerous antioxidant factors, such as HO-1 [19]. The results for Nrf-2 and HO-1 in the mouse livers are shown in Figure 6A,C,D. No significant difference in the performance of these two proteins was observed between the EtOH and NOR group (*p* > 0.05), but MS-L and AK-L significantly increased their expression levels. Nrf-2 expression was increased by 2.08 times (*p* < 0.001) and 2.26 times (*p* < 0.001), and HO-1 expression was increased by 3.34 times (*p* < 0.001) and 3.65 times (*p* < 0.001). According to these results, MS and AK increased antioxidant content and prevented oxidative stress damage by enhancing the transcription ability of Nrf-2 and HO-1. 

### 2.8. Effects of Monascin and Ankaflavin on The Regulation of Hepatic Lipid Metabolism Factor Expressions

p-AMPK is responsible for promoting fatty acid oxidation and for reducing fat formation, and it affects PPAR-α expression, which is responsible for fatty acid transportation [7]. Alcohol can hinder AMPK activation and enhance the production of both SREBP-1 and ACC, which in turn promotes fat production and accumulation in the liver. The results for liver AMPK and p-AMPK expression are presented in Figure 7A–C. No significant difference in AMPK was present among the groups but alcohol was observed to significantly reduce the p-AMPK/AMPK ratio compared with the NOR group. The ratio was decreased by 66.44% (*p* < 0.001). However, feeding low-dose MS or AK improved the alcohol-induced reduction of the p-AMPK/AMPK ratio; in the MS-L group, it was 1.51 times higher (*p* < 0.01), in the AK-L group 1.72 times higher (*p* < 0.001), and in the AK-H group 2.96 times higher (*p* < 0.001). Thus, MS and AK increased AMPK activation and promoted fatty acid oxidation. As can be observed in Figure 7A,D, alcohol significantly inhibited PPAR-α (by 55.28%, *p* < 0.001) but, relative to the EtOH group, its expression in the MS-L and AK-L group was 1.45 (*p* < 0.05) and 1.7 times (*p* < 0.01) higher, respectively. These results demonstrate that MS and AK recovered the p-AMPK/AMPK ratio that was decreased by alcohol in a dose-dependent manner.

To explore the effect of the test substances on liver lipid synthesis, we analyzed SREBP-1 and its downstream ACC expression. As Figure 7A,E,F indicates, a long-term alcohol diet increased SREBP-1 expression by 3.22 times (*p* < 0.001) and ACC expression was significantly higher than in the NOR group (2.6 times, *p* < 0.001), indicating that alcohol can promote liver fatty acid and TG synthesis. SREBP-1 expression was significantly reduced by 31.8% (*p* < 0.001) or 45.7% (*p* < 0.001) due to feeding low doses of MS or AK, and was reduced by 32.21% (*p* < 0.001) in the SL group. These results suggest that low-dose AK is more effective than SL in reducing SREBP-1 expression levels. In addition, the MS-L and AK-L groups exhibited ACC protein expression levels that were significantly reduced by 29.03% (*p* < 0.001) and 35.01% (*p* < 0.001), respectively. These results confirm that MS and AK reduced the synthesis of proteins related to TG and prevented the accumulation of TG in the liver.

## 3. Discussion

The induction mode adopted in the present study can successfully develop experimental animals with alcoholic fatty liver and hepatitis [20]. In the reports on the liver injury induced by the Lieber–DeCarli liquid feed, the AST and ALT of the alcohol liquid feed group were significantly higher than those of the normal group [21,22,23]. Therefore, feeding mice with the Lieber–DeCarli liquid diet was used to induce alcoholic liver injury and fatty liver in the current study. The results of the serum analysis demonstrated that the EtOH feed increased AST and ALT levels (Table 1), and eventually induced alcoholic liver injury in the mice.

Long-term alcohol abuse often causes obesity and malnutrition. The reason for these outcomes is that alcohol can produce 7.1 kcal/g. Through the ADH metabolic pathway, every mole of alcohol produces 16 moles of ATP [24,25]. In addition, when alcohol enters the body, the body consumes it as energy first and then stores it as fat. Alcohol directly damages the liver and intestines, reduces metabolism and absorption, and causes nutritional disorders [26]. In the present study, the body weight of the mice in the EtOH group was lower than that in the NOR group from the second week. However, the liver weight–body ratio of the EtOH group was higher than that of the NOR group. Nonetheless, MS and AK had significant effects in lowering the liver weight–body ratio (Figure 2).

Long-term alcohol intake induces CYP2E1 expression in alcohol metabolism, which further causes reactive oxygen species (ROS) generation, oxidative stress, and lipid peroxidation in liver cells. Lipid peroxidation hinders β-oxidation, promotes the expression of lipogenesis-related proteins, and finally causes liver damage [7,26,27]. The initial symptom of alcoholic liver injury is fatty liver. When TC and TG are continuously synthesized and their levels are increased, free fatty acids and chylomicrons more easily enter the liver, where they promote lipid production and inhibit lipolysis, leading to the accumulation of fat in the liver and inducing steatohepatitis risk [28,29]. In the results concerning lipid peroxidation, both MS and AK reduced the lipid peroxidation induced by the EtOH diet and the effect of AK was stronger than that of MS (Figure 4).

The results also indicated that the increasing trend of serum TC and TG content differed from that of the liver, which proves that TC and TG are formed by the metabolism of alcohol in the liver. In addition, the administration of MS and AK reduced TC and TG content in the liver. According to the results of the pathological section analysis, the tissue status of the MS and AK groups was similar to that of the normal group (Table 2). This result confirms that both MS and AK have the ability to prevent lipid accumulation in the liver and simultaneously prevent lipid peroxidation. Between them, AK was more effective than MS.

When the liver is damaged, a large amount of ROS, including H_2_O, ·O2^−^, ·OH, and ·ROO, are produced in liver cells [30]. If ROS cannot be effectively eliminated, inflammation ensues. Alcohol produces O_2_^−^ through the CYP2E1 pathway and both the SOD/CAT system and glutathione provide key antioxidation protection against alcohol-induced oxidative stress [31,32]. The results in Table 3 demonstrate that alcohol reduced the antioxidant capacity of the liver; however, MS and AK effectively enhanced the effects of SOD/CAT and GRd/GPx. Although both MS and AK are *Monascus*-produced yellow pigments, the effect of AK in enhancing the antioxidant system was still stronger than that of MS.

The main cause of alcoholic liver injury is the toxic effect of alcohol and its metabolite acetaldehyde, which promotes Kupffer cells to activate stellate cells, increases the expression of CYP2E1, and further stimulates liver inflammation through the formation of ROS and proinflammatory cytokines including TNF-α, IL-1β, IL-6, NF-κB, iNOS, and COX-2 [33,34,35]. In addition to the direct damage alcohol causes to the liver, it activates ERK1/2 and p38 MAPK in the liver by changing the permeability of the intestines, which causes proinflammatory factor expression and liver inflammation [34]. Our results (Figure 5) indicate that the expression levels of inflammatory cytokines TNF-α, IL-1β, and IL-6 in the EtOH group were increased compared to the NOR group. These results are consistent with those in the literature [36]. In addition, alcohol was reported to promote the phosphorylation of ERK1/2 and p38 MAPK, and to further increase NF-κB. The iNOS and COX-2 produced through NF-κB were also increased by alcohol. In our results, alcohol was observed to promote the expression of NF-κB downstream factors iNOS and COX-2. However, *Monascus*-produced MS and AK reversed the production of these inflammatory factors induced by alcohol in the liver. In addition, MS and AK inhibited the phosphorylation of ERK1/2 and p38 MAPK, thereby reducing the production of NF-κB and preventing both inflammation and damage caused by alcohol. MS and AK had potent effects at low doses and their anti-inflammatory effects at high doses were stronger than SL for many indicators, such as NF-κB, iNOS, and COX-2.

Past studies have demonstrated that PPAR-γ can regulate inflammation and inhibit the expression levels of NF-α, IL-1β, and IL-6 [37]. The results of the present study suggest that the expression level of PPAR-γ in the EtOH group was lower than that in the NOR group. This result is consistent with that of a previous study [18]. Our results also suggest that alcohol may cause inflammation not only by increasing inflammatory factors but also by inhibiting the expression of the PPAR-γ protein, causing more serious damage. However, decreased PPAR-γ expression caused by alcohol can be recovered by MS and AK ingestion (Figure 6). The oxidative stress caused by alcohol promotes the dissociation of transcriptional regulators Nrf-2 and Keap1, and enters the nucleus to initiate the transcription of antioxidant genes including HO-1, NQO-1, and GSTA-2 [38]. HO-1 maintains iron ion levels in cells and regulates antioxidant mechanisms. In past studies, HO-1 was a target factor for the treatment of liver diseases [39,40]. Our findings suggest that the expression levels of Nrf-2 and HO-1 in the EtOH group were not higher than those in the NOR group. When the body is producing excessive ROS, the activation of Nrf-2 is hindered and cell function is impaired, resulting in apoptosis and necrosis [19]. The inhibition of Nrf-2 activation affects downstream HO-1 performance. However, ingesting MS and AK can greatly increase Nrf-2 expression and the expression of its downstream factor HO-1 is also greatly increased, thus achieving antioxidant effects (Figure 6). A comprehensive comparison between the expression levels of proinflammatory factors and the liver function index confirmed that MS and AK can effectively prevent alcoholic liver injury. AK exhibited a superior improvement effect mainly because it inhibits TNF-α, IL-1β, and IL-6; enhances anti-inflammatory and antioxidant factors; and reduces NF-κB content by inhibiting p-p38 to reduce liver damage caused by alcohol.

Alcohol produces a large amount of NADH during metabolism and NADH increases GPDH activity as well as provides glycerol synthetic lipids. Therefore, ingesting alcohol promotes liver lipid biosynthesis [41]. AMPK and PPAR-α play the leading roles in lipid metabolism and biosynthesis in the liver. AMPK, a key energy regulator, promotes fatty acid oxidation, inhibits fatty acid synthesis, and inhibits the downstream expression of ACC, FAS, and GPAT by regulating SREBP-1 and SREBP-2, which finally inhibits fatty acid, cholesterol, and TG synthesis [42,43]. In addition, AMPK downregulates the expression of PPAR-α, which is regarded as a stimulator of the oxidation and transport of fatty acids. AMPK prevents the accumulation of TGs in the liver and reduces the apoptotic response caused by alcohol [7]. Previous studies have indicated that alcohol inhibits fatty acid oxidation and promotes lipogenesis in the liver [44]. As Figure 7 indicates, the EtOH group had a reduced p-AMPK/AMPK ratio; however, the administration of either MS or AK increased the AMPK expression lowered by alcohol. This study further confirmed that MS and AK can effectively inhibit the biosynthesis of TGs in the liver and prevent the formation of alcoholic fatty liver, as demonstrated by the results regarding SREBP-1 and ACC, as well as by the downstream regulation of the synthesis of TG. MS and AK increased the expression of PPAR-α, which was decreased by alcohol, and promoted fatty acid oxidation and transportation, thus reducing fatty acid accumulation in the liver.

MS and AK were proved to improve alcoholic fatty liver. The main regulatory mechanisms are as follows. MS and AK increase the expression of PPAR-α by stimulating AMPK phosphorylation, thereby promoting fatty acid β oxidation. In addition, MS and AK inhibit the expression of SREBP-1 by increasing AMPK phosphorylation, further inhibiting the expression of ACC to restrict the fatty acid biosynthesis pathway. These two effects reduce the formation and accumulation of liver TG, thereby achieving the effect of improving alcoholic fatty liver. AK increased AMPK phosphorylation and PPAR-α expression more than MS, and thus its ability to reduce liver TGs and prevent alcoholic fatty liver is superior.

Although MS and AK have significant effects on the regulation of multiple fatty liver and inflammatory factors, the high-dose and low-dose groups cannot show a dose–response relationship in some results, such as that of the antioxidative system enzymes and Nrf-2. The reason may be that the test substance had reached the maximum effect on these regulatory factors at low doses. Therefore, even if the dose is increased, its effect cannot be significantly improved. In addition, some results, such as that of the MDA content of liver lipid peroxidation, also showed that the results of the low-dose MS and AK groups were close to that of the normal group; therefore, increasing the dose of MS and AK experienced difficulty increasing the effect. They may reach the maximum effect.

We conclude that the multiple regulatory mechanisms of MS and AK in the prevention of alcoholic liver injury include the regulation of lipid oxidation stress, inflammation, and lipid biosynthesis pathways. As can be seen in Figure 8, MS and AK prevent oxidative stress by increasing the activity of the antioxidant enzymes CAT, SOD, GPx, and GRd, and by reducing lipid peroxidation. In addition, MS and AK inhibit the phosphorylation of the MAPK family, including the phosphorylation of p38 and ERK1/2, to enhance the regulation of NF-κB and its downstream oxidative inflammatory factors including TNF-α, IL-1β, IL-6, iNOS, and COX-2. In addition, the expression of anti-inflammatory factors PPAR-γ, Nrf-2, and HO-1 was also increased by the regulation promoted by MS and AK. In regulating lipid biosynthesis pathways, MS and AK mainly increased the expression of phosphorylated AMPK and PPAR-α, thereby inhibiting fatty acid synthesis and promoting β-oxidation pathways. This study confirmed that MS and AK can improve the effects of alcohol-induced fatty liver and liver damage through a combined mechanism.

## 4. Materials and Methods

### 4.1. Chemicals

Monascin and ankaflavin (purities > 99.9%), provided by SunWay Biotech Co., LTD (Taipei, Taiwan), were extracted and purified from Monascus purpureus NTU 568-fermented rice by following the methods of previous research [45]. Silymarin (S0292) was purchased from Sigma Chemical Co., LTD (St. Louis, MO, USA). Monoclonal ACC1 antibody (3676), polyclonal AMPKα antibody (2532), monoclonal p-AMPKα antibody (2535), monoclonal ERK1/2 antibody (4695), monoclonal p-ERK1/2 antibody (4370), polyclonal p38-MPAK antibody (9212), and polyclonal p-p38 antibody (9211) were purchased from Cell Signaling Technology, Inc. (Danvers, MA, USA). Polyclonal Nrf2 antibody (ab31163) and monoclonal HO-1 antibody (ab13248) were purchased from Abcam (Cambridge, UK). Polyclonal IL-6 antibody (sc-1265), polyclonal IL-1β antibody (sc-7884), monoclonal NF-κB antibody (sc-8008), polyclonal PPARα antibody (sc-9000), polyclonal SREBP-1 antibody (sc-8984), and polyclonal COX-2 antibody (sc-1747) were purchased from Santa Cruz Biotechnology, Inc. (Dallas, TX, USA). Monoclonal β-actin antibody (MA5-15739) and polyclonal iNOS antibody (PA3-030A) were purchased from Thermo Fisher Scientific Inc. (Rockford, IL, USA). Polyclonal PPARγ antibody (07-466) and polyclonal TNF-α antibody (AB2148P) were purchased from Millipore (Darmstadt, Germany)

### 4.2. Animal Models and Grouping

Fifty-six male C57BL/6J mice (6 weeks old) purchased from the National Laboratory Animal Center (Taipei, Taiwan) were kept in plastic cages and underwent a 12 h light/dark cycle at 60% relative humidity and 23 °C. Animals could obtain regular rodent food and water for free. After two weeks of acclimation, the mice were weighed and randomly divided into 7 groups with 8 mice in each group.

Both MS and AK are the main functional components in *Monascus purpureus*-fermented red mold rice. According to past studies, the concentrations of MS and AK in red mold rice are 3 mg/kg and 1.5 mg/kg. The recommended dosage of red mold rice was 1 g, which included 3 mg of MS and 1.5 mg of AK. Past studies have also emphasized that this dose has the function of regulating blood lipids. Therefore, this study used 3 mg/kg b.w. as the low dose of the test substance for an adult and set 5 times this dose as the high dose. The feeding dose algorithm was based on the method of estimating the maximum safe starting dose in initial clinical trials for therapeutics in adult healthy volunteers, as announced by the U.S. Food and Drug Administration in 2005 [46]. According to the dose conversion formula (mouse dose (mg/kg bw) = daily adult dose (g)/60 kg*12.3), the low dose of MS and AK is 0.615 and 0.3075 mg/kg b.w./day, respectively. Silymarin is a supplement and an alternative medicine. It has antioxidant, anti-inflammatory, and anti-fibrotic effects for the prevention of liver diseases. Past studies have also indicated that it has functions for improving the oxidative stress, inflammation, and steatosis caused by an alcoholic diet [47]. Therefore, silymarin was used as the test substance in the positive control group in this study.

The mice of the NOR group were fed the Lieber-DeCarli Regular Control Diet (DYET# 710027) and given RO water orally. The other mice had alcoholic liver injury induced by being fed daily the Lieber-DeCarli Ethanol diet (DYET#710260). Meanwhile, the mice were orally administered RO water (EtOH group), silymarin (200 mg/kg bw/day), a low dose of monascin (0.615 mg/kg bw/day; MS-L group), a high dose of monascin (3.075 mg/kg bw/day; MS-H group), ankaflavin (0.3075 mg/kg bw/day; AK-L group), or a high dose of ankaflavin (1.5375 mg/kg bw day; AK-H group). The experiment was reviewed and approved (code: 1050401) by the Animal Care and Research Ethics Committee of the National Taitung University.

### 4.3. Serum Biochemistry Parameters

After animal test for six weeks, the mice were deprived of food for 16 h before being scarified by CO_2_ asphyxiation. Blood samples were collected from the posterior vena cava and centrifuged at 5000× *g* for 10 min; the serum was stored at −20 °C until analyzed. Aspartate aminotransferase (AST), alanine aminotransferase (ALT), and alkaline phosphatase (ALP) in serum were measured using the automated clinical chemistry analyzer (Beckman-700, Fullerton, CA, USA). The levels of serum lipids were determined by using the triglyceride (TG) assay kit and total cholesterol (TC) assay kit (BXC0271 and BXC0261, Fortress Diagnostics Limited, Antrim, UK).

### 4.4. Liver Biochemistry Parameters

The part of the largest lobe of liver tissue was immersed in 10% formaldehyde for histological examination. Other liver tissues were stored at −80 °C for further analysis. The liver homogenates with lysis buffer were used to evaluate lipid peroxidation by the thiobarbituric acid reactive substance (TBARS) assay [48].

For the liver lipid extraction, the liver tissue (0.1 g) was ground in 1 mL of ice-cold Folch solution (chloroform/methanol = 2:1; *v*/*v*) and incubated for 30 min at room temperature. The aqueous layer was aspirated and discarded, and the fixed volume of the organic layer was then evaporated to dryness. The dried lipid layer was dissolved with an equal volume of DMSO and then used to determine the TC (BXC0261, Fortress Diagnostics Ltd.) and TG (BXC0271, Fortress Diagnostics Ltd.) levels using a commercial assay kit. The cholesterol and triglyceride were used as standards for the standard curve of the TC and TG analysis, respectively. Liver homogenates with DMSO were used to determine liver lipids by using the ELISA reader (Multiskan™ GO, Thermo Fisher Scientific Inc., Waltham, MA, USA).

Regarding the analysis of liver antioxidative enzymes activity, the liver tissue was ground in ice-cold phosphate-buffered saline (PBS, 0.01M, pH = 7.4) and then centrifuged (5000× *g*, 5 min). The liver homogenates were measured according to the activities of anti-oxidative enzymes by using the SOD assay kit (SD 125, Randox, UK), CAT assay kit (EnzyChromTM Catalase Assay Kit, ECAT-100, BioAssay Systems, Hayward, CA, USA), GPx assay kit (RS 505, Randox, UK), and GRd assay kit (GR 2368, Randox, UK).

### 4.5. Immunoblotting

The liver tissue was homogenized in lysis buffer (1% Triton X-100, 20 mM Tris, 40 mM NaF, 0.2% SDS, 0.5% deoxycholate, 1 mM EDTA, 1 mM EGTA, 1 mM Na_3_VO_4_, and 100 mM NaCl, pH 7.5). The extract was centrifuged at 15,000× *g* for 15 min at 4 °C and then stored at −20 °C before use. The target protein in the liver tissue sample was applied for the immunoblotting according to previous studies [49,50]. The bicinchoninic acid (BCA) protein assay kit (23225, Thermo Fisher Scientific Inc., Rockford, IL, USA) was used to measure the total protein concentration.

The samples were separated on 10% SDS-PAGE gels and then transferred to polyvinylidene fluoride membranes. The blots were incubated with various primary antibodies at room temperature for 1 h. The protein bands were then incubated with horseradish peroxidase (HRP)-conjugated secondary antibodies at room temperature for 1 h and visualized by the enhanced chemiluminescence (ECL) substrate with the UVP AutoChemi Image system (UVP Inc., Upland, CA, USA). Immunoblotting band intensities were quantified using UVP LabWork 4.5 software (UVP Inc.). The protein expressions were detected by immunoblotting analysis. The levels of the proteins were subsequently quantitated by densitometric analysis, with that of the control being 100%.

### 4.6. Histological Stains

Fixed-liver tissue was cut into 7 μm thickness and mounted on silanized slides (Dako Japan, Tokyo, Japan). The hematoxylin and eosin (H&E)-staining was done as the principal staining in the liver slice for the histopathological examination.

### 4.7. Statistics

Data are expressed as means ± standard deviation. The statistical analysis was determined by one-way analysis of variance (ANOVA) with Duncan’s multiple range test of SPSS 12.0 (SPSS Institute, Inc., Chicago, IL, USA). Differences with *p* < 0.05 were considered statistically significant.

## Figures and Tables

**Figure 1 molecules-26-06301-f001:**
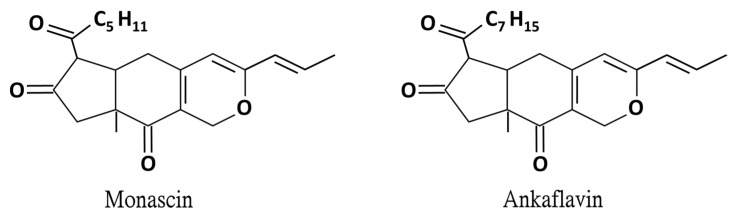
Chemical structure of monascin and ankaflavin.

**Figure 2 molecules-26-06301-f002:**
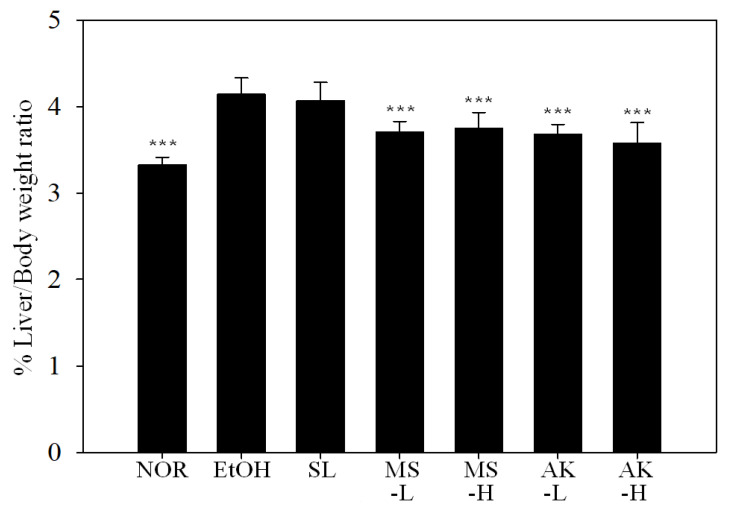
Liver weight–body ratio of the mice fed with the liquid Lieber—DeCarli ethanol diet. Two groups of the mice were fed a normal diet (NOR group) or a liquid Lieber—DeCarli ethanol diet (EtOH group) without the administration of test materials. The other ethanol-induced liver injury mice were administered silymarin (200 mg/kg bw/day), a low dose of monascin (0.615 mg/kg bw/day; MS-L group), a high dose of monascin (3.075 mg/kg bw/day; MS-H group), ankaflavin (0.3075 mg/kg bw/day; AK-L group), or a high dose of ankaflavin (1.5375 mg/kg bw day) (AK-H group). Data are presented as mean ± SD (*n* = 8). *** *p* < 0.001 vs. EtOH group.

**Figure 3 molecules-26-06301-f003:**
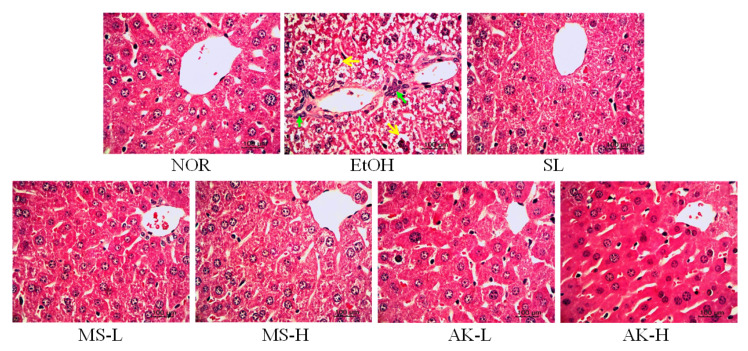
Hepatic pathological changes in the mice fed with the liquid Lieber–DeCarli ethanol diet (magnification 400×; the arrows indicate the position of liver injury). Two groups of the mice were fed a normal diet (NOR group) or a liquid Lieber–DeCarli ethanol diet (EtOH group) without the administration of test materials. The other ethanol-induced liver injury mice were administered silymarin (200 mg/kg bw/day), a low dose of monascin (0.615 mg/kg bw/day; MS-L group), a high dose of monascin (3.075 mg/kg bw/day; MS-H group), ankaflavin (0.3075 mg/kg bw/day; AK-L group), or a high dose of ankaflavin (1.5375 mg/kg bw day; AK-H group).

**Figure 4 molecules-26-06301-f004:**
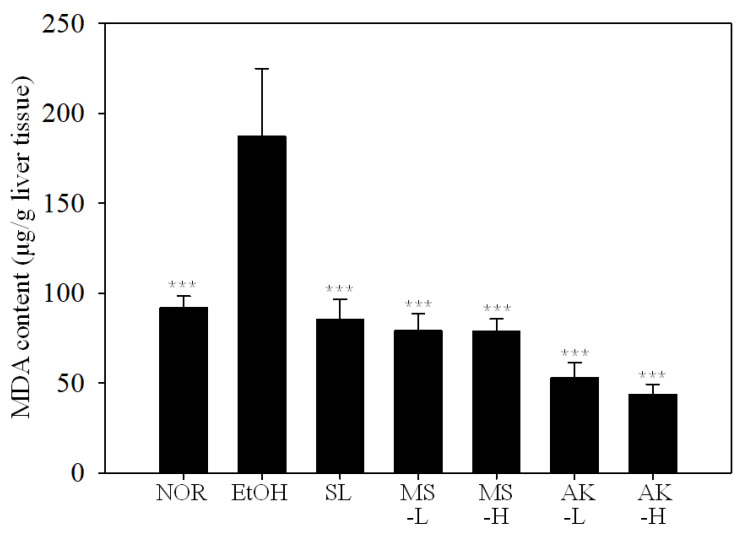
Liver lipid peroxidation in the mice fed with the liquid Lieber–DeCarli ethanol diet. Two groups of the mice were fed a normal diet (NOR group) or a liquid Lieber–DeCarli ethanol diet (EtOH group) without the administration of test materials. The other ethanol-induced liver injury mice were administered silymarin (200 mg/kg bw/day), a low dose of monascin (0.615 mg/kg bw/day; MS-L group), a high dose of monascin (3.075 mg/kg bw/day; MS-H group), ankaflavin (0.3075 mg/kg bw/day; AK-L group), or a high dose of ankaflavin (1.5375 mg/kg bw day; AK-H group). Data are presented as mean ± SD (*n* = 8). *** *p* < 0.001 vs. EtOH group. Abbreviation: MDA, malondialdehyde.

**Figure 5 molecules-26-06301-f005:**
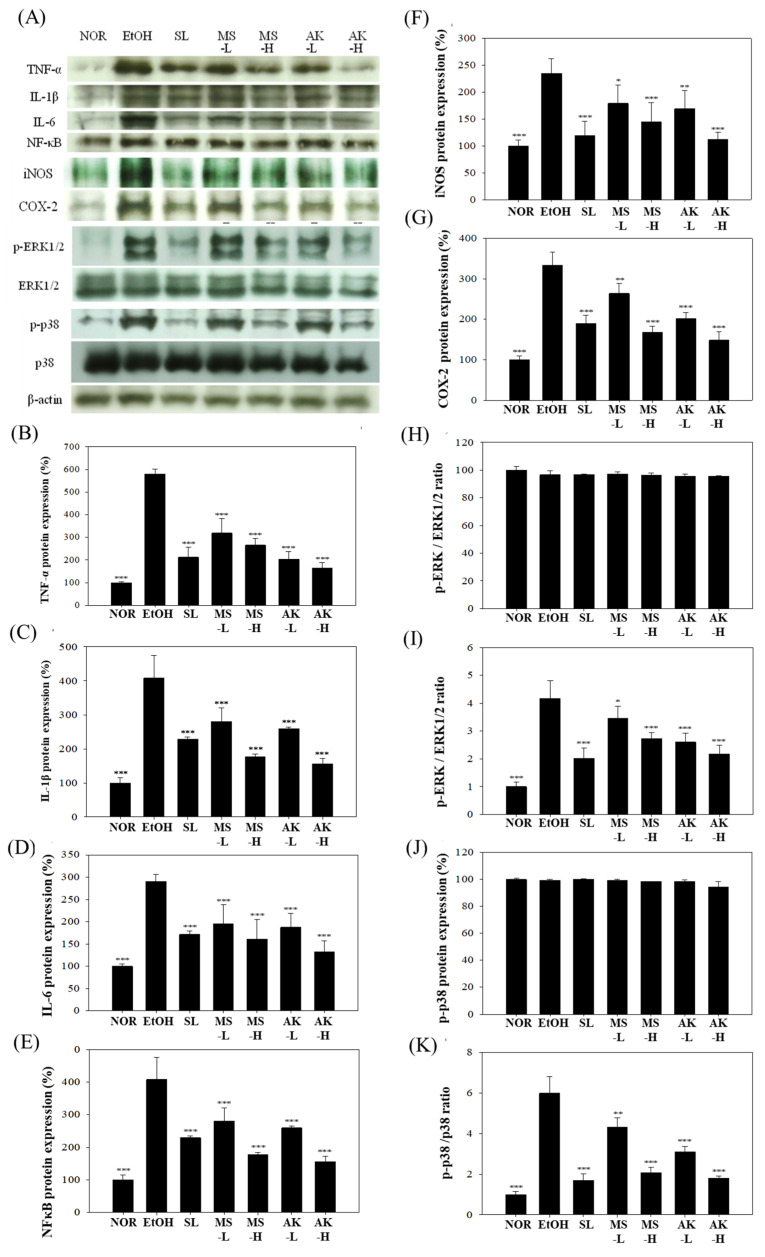
Hepatic inflammatory factor expressions in the mice fed with the liquid Lieber–DeCarli ethanol diet. (**A**) Immunoblotting photograph of the (**B**) TNF-α, (**C**) IL-1β, (**D**) IL-6, (**E**) NF-κB, (**F**) iNOS, (**G**) COX-2, (**H**) ERK1/2, (**I**) p-ERK/ERK1/2 ratio, (**J**) p38, and (**K**) p-p38/p38 ratio. Two groups of the mice were fed a normal diet (NOR group) or a liquid Lieber–DeCarli ethanol diet (EtOH group) without the administration of test materials. The other ethanol-induced liver injury mice were administered silymarin (200 mg/kg bw/day), a low dose of monascin (0.615 mg/kg bw/day; MS-L group), a high dose of monascin (3.075 mg/kg bw/day; MS-H group), ankaflavin (0.3075 mg/kg bw/day; AK-L group), or a high dose of ankaflavin (1.5375 mg/kg bw day; AK-H group). Data are presented as mean ± SD (*n* = 8). * *p* < 0.05, ** *p* < 0.01, and *** *p* < 0.001 vs. EtOH group. Abbreviations: TNF-α, tumor Necrosis Factor-α; IL-1β, Interleukin 1 beta; IL-6, Interleukin 6; NF-κB, nuclear factor kappa-light-chain-enhancer of activated B; iNOS, inducible nitric oxide synthase; COX-2, cyclooxygenase-2; and ERK1/2, extracellular signal-regulated kinase 1/2.

**Figure 6 molecules-26-06301-f006:**
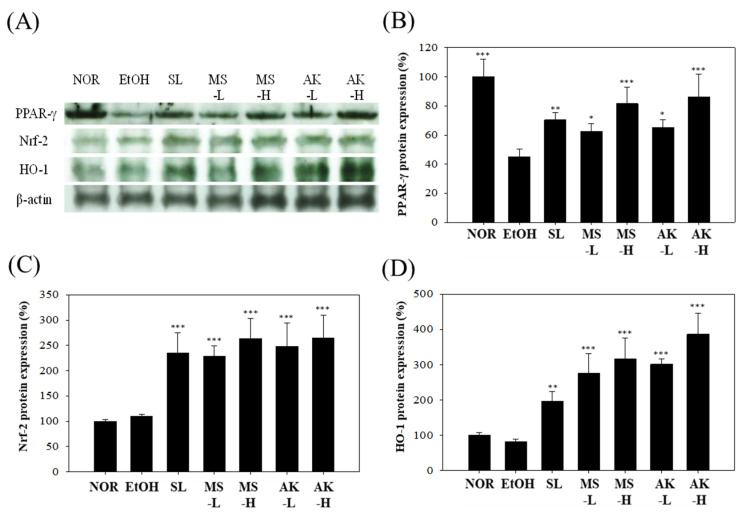
Hepatic anti-inflammatory factor expressions in the mice fed with the liquid Lieber–DeCarli ethanol diet. (**A**) Immunoblotting photograph of (**B**) PPAR-γ, (**C**) Nrf-2, and (**D**) HO-1. Two groups of the mice were fed a normal diet (NOR group) or a liquid Lieber–DeCarli ethanol diet (EtOH group) without the administration of test materials. The other ethanol-induced liver injury mice were administered silymarin (200 mg/kg bw/day), a low dose of monascin (0.615 mg/kg bw/day; MS-L group), a high dose of monascin (3.075 mg/kg bw/day; MS-H group), ankaflavin (0.3075 mg/kg bw/day; AK-L group), or a high dose of ankaflavin (1.5375 mg/kg bw day) (AK-H group). Data are presented as mean ± SD (*n* = 8). * *p* < 0.05, ** *p* < 0.01, and *** *p* < 0.001 vs. EtOH group. Abbreviations: PPAR-γ, proliferator-activated receptor γ; Nrf-2, nuclear factor erythroid 2–related factor 2; and HO-1, heme oxygenase-1.

**Figure 7 molecules-26-06301-f007:**
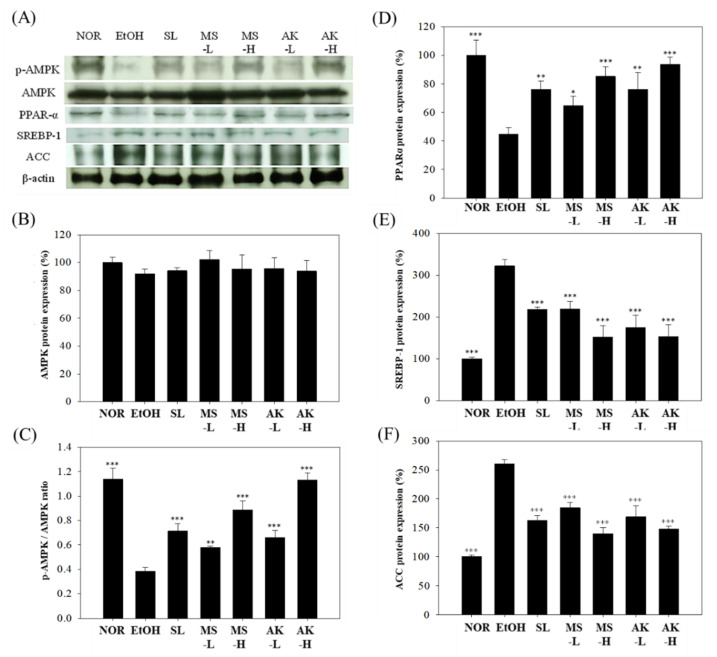
Hepatic lipid metabolism factor expressions in the mice fed with the liquid Lieber–DeCarli ethanol diet. (**A**) Immunoblotting photograph of the (**B**) AMPK and (**C**) p-AMPK/AMPK ratio, and (**D**)PPAR-α, (**E**) SREBP-1, and (**F**) ACC. Two groups of the mice were fed a normal diet (NOR group) or a liquid Lieber–DeCarli ethanol diet (EtOH group) without the administration of test materials. The other ethanol-induced liver injury mice were administered silymarin (200 mg/kg bw/day), a low dose of monascin (0.615 mg/kg bw/day; MS-L group), a high dose of monascin (3.075 mg/kg bw/day; MS-H group), ankaflavin (0.3075 mg/kg bw/day; AK-L group), or a high dose of ankaflavin (1.5375 mg/kg bw day; AK-H group). Data are presented as mean ± SD (*n* = 8). * *p* < 0.05, ** *p* < 0.01, and *** *p* < 0.001 vs. EtOH group. Abbreviations: AMPK, AMP-activated protein kinase; PPAR-α, proliferator-activated receptor-α; SREBP-1, sterol regulatory element-binding protein-1; and ACC, acetyl-Coenzyme A carboxylase.

**Figure 8 molecules-26-06301-f008:**
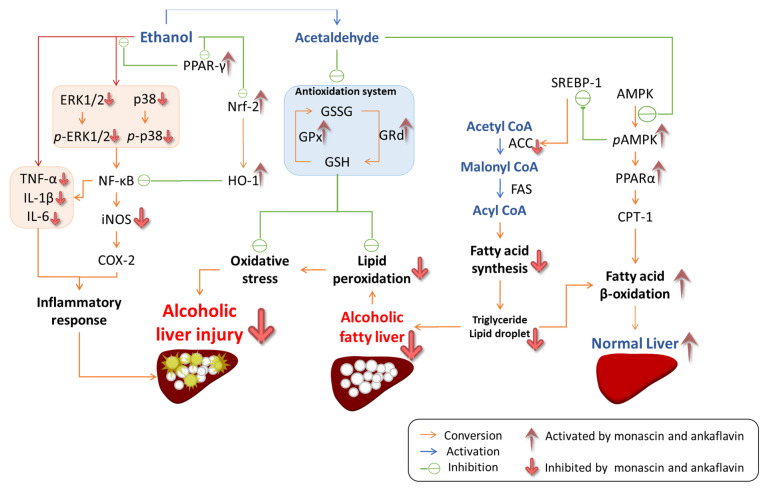
The regulations of monascin and ankaflavin in terms of the prevention of alcoholic liver injury and fatty liver in the mice fed with the liquid Lieber–DeCarli ethanol diet.

**Table 1 molecules-26-06301-t001:** Serum AST, ALT, and ALP activities in the mice fed with the liquid Lieber–DeCarli ethanol diet.

Groups	AST (U/L)	ALT (U/L)	ALP (IU/L)
NOR	44.9 ± 4.9 *	17.3 ± 2.5 **	67.0 ± 3.1 **
EtOH	51.0 ± 9.8	22.1 ± 6.2	76.0 ± 5.5
SL	43.0 ± 4.7 **	18.5 ± 2.4 *	73.3 ± 5.9
MS-L	41.6 ± 3.7 **	16.8 ± 2.7 **	70.6 ± 4.1
MS-H	39.5 ± 3.4 ***	16.3 ± 1.9 **	68.6 ± 6.1 *
AK-L	42.1 ± 4.3 **	17.0 ± 2.8 **	68.1 ± 4.3 **
AK-H	38.9 ± 3.2 ***	16.8 ± 0.5 **	69.5 ± 6.7 *

Two groups of the mice were fed a normal diet (NOR group) or a liquid Lieber—DeCarli ethanol diet (EtOH group) without the administration of test materials. The other ethanol-induced liver injury mice were administered silymarin (200 mg/kg bw/day), a low dose of monascin (0.615 mg/kg bw/day; MS-L group), a high dose of monascin (3.075 mg/kg bw/day; MS-H group), ankaflavin (0.3075 mg/kg bw/day; AK-L group), or a high dose of ankaflavin (1.5375 mg/kg bw day; AK-H group). Data are presented as mean ± SD (*n* = 8). * *p* < 0.05, ** *p* < 0.01, and *** *p* < 0.001 vs. EtOH group.

**Table 2 molecules-26-06301-t002:** Serum or hepatic triglyceride and cholesterol levels in the mice fed with the liquid Lieber–DeCarli ethanol diet.

Groups	Serum		Liver	
	TC (mg/dL)	TG (mg/dL)	TC (mg/g)	TG (mg/g)
NOR	65.62 ± 4.73	99.61 ± 10.39 **	3.41 ± 0.05 ***	3.10 ± 0.56 ***
EtOH	68.82 ± 10.40	119.61 ± 14.32	3.77 ± 0.16	4.86 ± 0.84
SL	63.74 ± 5.11	110.33 ± 8.78 *	3.43 ± 0.14 ***	3.01 ± 0.39 ***
MS-L	58.35 ± 7.39 *	94.85 ± 10.50 **	3.43 ± 0.11 ***	2.95 ± 0.38 ***
MS-H	60.72 ± 7.09 *	88.45 ± 4.89 ***	3.19 ± 0.08 ***	2.52 ± 0.21 ***
AK-L	59.32 ± 4.45 *	106.48 ± 8.48 *	3.22 ± 0.15 ***	2.85 ± 0.53 ***
AK-H	60.49 ± 5.72 *	96.88 ± 9.56 **	3.10 ± 0.09 ***	2.44 ± 0.18 ***

Two groups of the mice were fed a normal diet (NOR group) or a liquid Lieber—DeCarli ethanol diet (EtOH group) without the administration of test materials. The other ethanol-induced liver injury mice were administered silymarin (200 mg/kg bw/day), a low dose of monascin (0.615 mg/kg bw/day; MS-L group), a high dose of monascin (3.075 mg/kg bw/day; MS-H group), ankaflavin (0.3075 mg/kg bw/day; AK-L group), or a high dose of ankaflavin (1.5375 mg/kg bw day; AK-H group). Data are presented as mean ± SD (*n* = 8). * *p* < 0.05, ** *p* < 0.01, and *** *p* < 0.001 vs. EtOH group. Abbreviations: TC, total cholesterol and TG, triglyceride.

**Table 3 molecules-26-06301-t003:** Antioxidant enzyme activities in the mice fed with the liquid Lieber–DeCarli ethanol diet.

Groups	CAT Activity (U/mg Protein)	SOD Activity (U/mg Protein)	GPx Activity (U/mg Protein)	GRd Activity (U/mg Protein)
NOR	0.54 ± 0.06 ***	4.43 ± 0.65 ***	807.02 ± 60.02 ***	68.67 ± 6.02 ***
EtOH	0.42 ± 0.07	2.59 ± 0.23	611.87 ± 53.06	57.76 ± 3.84
SL	0.43 ± 0.08	3.55 ± 0.52 **	666.49 ± 52.95	65.11 ± 4.25 **
MS-L	0.62 ± 0.03 ***	3.12 ± 0.35	604.53 ± 64.00	63.31 ± 5.92 *
MS-H	0.56 ± 0.06 ***	3.60 ± 0.61 **	678.79 ± 52.78 *	71.39 ± 4.82 ***
AK-L	0.54 ± 0.04 ***	3.36 ± 0.63 **	653.51 ± 37.76	66.05 ± 7.38 **
AK-H	0.58 ± 0.07 ***	3.82 ± 0.68 ***	685.66 ± 71.14 *	73.18 ± 3.35 ***

Two groups of the mice were fed a normal diet (NOR group) or a liquid Lieber–DeCarli ethanol diet (EtOH group) without the administration of test materials. The other ethanol-induced liver injury mice were administered silymarin (200 mg/kg bw/day), a low dose of monascin (0.615 mg/kg bw/day; MS-L group), a high dose of monascin (3.075 mg/kg bw/day; MS-H group), ankaflavin (0.3075 mg/kg bw/day; AK-L group), or a high dose of ankaflavin (1.5375 mg/kg bw day; AK-H group). Data are presented as mean ± SD (*n* = 8). * *p* < 0.05, ** *p* < 0.01, and *** *p* < 0.001 vs. EtOH group. Abbreviations: GPx, glutathione peroxidase; GRd, glutathione reductase; CAT, catalase; and SOD, superoxide dismutase.

## Data Availability

The data presented in this study are available on request from the corresponding author. The data are not publicly available due to ethical restriction.

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
