# Peer review of "Monascin and Ankaflavin of Monascus purpureus Prevent Alcoholic Liver Disease through Regulating AMPK-Mediated Lipid Metabolism and Enhancing Both Anti-Inflammatory and Anti-Oxidative Systems"

_molecules, 2021, doi:10.3390/molecules26206301_

Round 1

Reviewer 1 Report

The authors present a very detailed characterization of the action mechanism of how monascin and ankaflavin of Monascus purpureus preventing liver disease. The manuscript is well structured and will be of interest to those working with liver injury and fatty liver. I recommend its acceptance for publication after minor details

1.- In figure 2, the last sentence should be deleted, it does not correspond with the figure.

2.- In the experiments there is no grater difference in the effect between the lower and higher concentrations of each compound, Is there an explanation about this? It should be desirable to include this in the manuscript.

Author Response

The authors present a very detailed characterization of the action mechanism of how monascin and ankaflavin of Monascus purpureus preventing liver disease. The manuscript is well structured and will be of interest to those working with liver injury and fatty liver. I recommend its acceptance for publication after minor details

1.- In figure 2, the last sentence should be deleted, it does not correspond with the figure.

Response: Thanks for your review. We have deleted this sentence. 

2.- In the experiments there is no grater difference in the effect between the lower and higher concentrations of each compound, Is there an explanation about this? It should be desirable to include this in the manuscript.

Response: Thanks for your review. Although MS and AK have significant effects on the regulation of multiple fatty liver and inflammatory factors, the high-dose and low-dose groups cannot show a dose-response relationship in some results such as antioxidative system enzymes, and Nrf-2). The reason may be that the test substance has reached the maximum effect on these regulatory factors at low doses. Therefore, even if the dose is increased, its effect cannot be significantly improved. In addition, some results such as MDA content of liver lipid peroxidation also showed that the results of the low dose MS and AK groups were close to that of the normal group, therefore, increasing the dose of MS and AK was difficult to increase the effect. They may reach the maximum effect. (Line 458-466)

Reviewer 2 Report

The paper deals with the effects of two compounds from red rice mould, widely used in the production of fermented rice. Their effects on alcoholic liver disease is studied. The effects are compared with another well-known plant extract called silymarin. The paper is well written in good English and the study is quite detailed.  The paper can be published with minor corrections:

  1. In multiple legends to Figures and tables si-lymarin is written. Change it to silymarin.
  2. Opponent recommends  showing the structures of the compounds in question.
  3. Include DOI  where appropriate in references.

Author Response

The paper deals with the effects of two compounds from red rice mould, widely used in the production of fermented rice. Their effects on alcoholic liver disease is studied. The effects are compared with another well-known plant extract called silymarin. The paper is well written in good English and the study is quite detailed.  The paper can be published with minor corrections:

  1. In multiple legends to Figures and tables si-lymarin is written. Change it to silymarin.

Response: Thanks for your review. We have revised the mistakes.

  1. Opponent recommends showing the structures of the compounds in question.

Response: Thanks for your review. We have added the chemical structure in the Fig. 1 (Line 75-79).

  1. Include DOI where appropriate in references.

Response: Thanks for your review. We have added the DOI in the reference.

Reviewer 3 Report

In the present study, yellow Monascus pigments, monascin (MS) and ankaflavin (AK), were used to prevent liver disease caused by alcohol. Alcoholic liver injury could be prevented by MS and AK by regulating lipid oxidation stress, inflammation and lipid biosynthesis pathways. It is an interesting topic in the related areas, but the manuscript needs some improvement before acceptance for publication.Some comments are as follow:

  1. Page 2, line 80 Why did you choose silymarin as positive control? Please explain and provide the references.
  2. Page 2, line 81-82  What are the rationales of the low and high dosage of MS and AK?
  3. Page 15, line 480  Please give the purities of MS and AK.
  4. Page 17, line 599  The page number is missed.

Author Response

In the present study, yellow Monascus pigments, monascin (MS) and ankaflavin (AK), were used to prevent liver disease caused by alcohol. Alcoholic liver injury could be prevented by MS and AK by regulating lipid oxidation stress, inflammation and lipid biosynthesis pathways. It is an interesting topic in the related areas, but the manuscript needs some improvement before acceptance for publication. Some comments are as follow:

1. Page 2, line 80 Why did you choose silymarin as positive control? Please explain and provide the references.

Response: Thanks for your review. Silymarin is a supplement and alternative medicine. It has antioxidant, anti-inflammatory, anti-fibrotic effects for the prevention of liver diseases. Past studies have also indicated that it has the functions of improving oxidative stress, inflammation, and steatosis caused by alcoholic diet. Therefore, silymarin was used as the test substance in the positive control group in this study. We have added the description and reference in the revised manuscripts (Line 517-522).

2. Page 2, line 81-82  What are the rationales of the low and high dosage of MS and AK?

Response: Thanks for your review. Both MS and AK are the main functional components in Monascus purpureus-fermented red mold rice. According to past studies, the concentrations of MS and AK in red mold rice are 3 mg/kg and 1.5 mg/kg. the recommendation dosage of red mold rice was 1 g, which included 3 mg of MS and 1.5 mg of AK. Past studies have also pointed out that this dose has the function of regulating blood lipids. Therefore, this study used 3 mg/kg b.w. as the low dose of the test substance of an adult, and set 5 times the dose as the high dose. The feeding dose algorithm is based on the method of Estimating the maximum safe starting dose in initial clinical trials for therapeutics in adult healthy volunteers announced by the U.S. Food and Drug Administration in 2005 [46]. According to the dose conversion formula (mouse dose (mg/kg bw) = daily adult dose (g) / 60 kg*12.3), the low dose of MS and AK is 0.615 and 0.3075 mg/kg b.w./day, respectively  (Line 507-517).

3. Page 15, line 480  Please give the purities of MS and AK.

Response: Thanks for your review. We have added the purities in the revised manuscript (Line 484)

4. Page 17, line 599  The page number is missed.

Response: Thanks for your review. We have added the purities in the revised manuscript (Line 484)